# Impacts of Tide Gate Modulation on Ammonia Transport in a Semi-closed Estuary during the Dry Season—A Case Study at the Lianjiang River in South China



**Changjin Zhao [1], Hanjie Yang [1], Zhongya Fan [1,\*] , Lei Zhu [2], Wencai Wang [1] and Fantang Zeng [1]**

[1] National Key Laboratory of Water Environmental Simulation and Pollution Control, Guangdong Key Laboratory of Water and Air Pollution Control, South China Institute of Environmental Sciences, Ministry of Ecology and Environment, No. 18 Ruihe Road, Guangzhou 510530, China; zhaochangjin@scies.org (C.Z.); yanghanjie@scies.org (H.Y.); wangwencai@scies.org (W.W.); zengfantang@scies.org (F.Z.)

[2] School of Marine Science, Sun Yat-sen University, 135 Xingangxi Rd., Guangzhou 510275, China; zhulei28@mail.sysu.edu.cn

\* Correspondence: fanzhongya@scies.org; Tel.: +020-85544702

**Abstract:** Recovery of tide-receiving is considered to improve the water quality in the Lianjiang River, a severely polluted and tide-influenced river connected to the South China Sea. A tide-receiving scenario, i.e., keeping the tide gate open, is compared with the other scenario representing the non-tide-receiving condition, i.e., blocking the tide flow during the flood phase, by numerical simulations based on the EFDC (Environmental Fluid Dynamics Code) model. The impacts of tide receiving were evaluated by the variation in the concentration of ammonia and its exporting fluxes, mainly in the downstream part of the river. With more water mass coming into the river, in the tide-receiving scenario, the averaged concentration of ammonia reduced by 20–40%, with the most significant decrease of 0.64 g m$^{-3}$. However, the exporting flux of ammonia has decreased in the tide-receiving scenario, as the consequence of the back–forth oscillation of tidal current. In the tide-receiving scenario, the time series of ammonia concentration approximately followed the tidal oscillation, with increased concentration during the ebb tide and reduction in the flood tide. In the non-tide-receiving scenario, the ammonia concentration decreases when the tide gate is open which results in further intrusion of seawater. This was followed by an increase in ammonia concentration again after the currents shift seaward and water mass with higher concentration from the upstream part is transported downstream. Given the identical ammonia input and river runoff, the ammonia concentration stays lower in the tide-receiving scenario, except for short periods after the tide gate opening and neap tides in the downstream part which lasts for around half a day. This study highlights the importance of hydrodynamic condition, specifically tidal oscillation, in the semi-diurnal and fortnight cycles, for the transportation of waterborne materials. Furthermore, the operation of the tide gate was additionally discussed based on potential varied practical conditions and evaluation criteria.

**Keywords:** tide gate; ammonia transport; tide-influenced river; estuary

## 1. Introduction

The highly urbanized estuary and coastal areas, due to its high industrialization, with dense popularity and variability of near-coastal systems, are susceptible to multiple pressures induced by anthropological influence [1–3]. In the last century, the increased demand for agricultural irrigation,

limited availability of freshwater, and the requirement to protect lands from inundation and salty water intrusion have initiated the building of water infrastructures, such as tide gates, worldwide [4]. Tide gates are generally doors or flaps mounted on the downstream ends of rivers near the sea, which allow upstream waters to drain while preventing inflows from the sea due to tidal surges or flood events [5,6]. However, tide gate could disrupt the natural ecosystem by reducing the water connectivity between the river and sea, which potentially would result in undesirable physical, chemical, and biological side effects, such as low dissolved oxygen, high nutrients concentration and intensified algae bloom [7,8], blocked or delayed fish passage [9], invasion of upland plant, and so on. Additionally, in some areas where local industries have experienced rapid development, the degraded water connectivity could further deteriorate undesirable water quality [10]. In recent decades, easing the conflicts between economic development and environmental capacity is not only the urgent desire of public members to improve life quality, but also an important guarantee for cities and promoting sustainable economic and social development [11]. Taking account of the pros and cons of coastal infrastructure, how to utilize or restore the previously constructed tide gate to adapt to the new requirements of economic and environmentally friendly development, has drawn increasing attention and been under fierce debate [12].

To improve the flushing of tide-gate-controlled rivers has long been recognized as an important part of solving the problem of undesired water quality in poorly water-connected rivers [13,14]. This is precisely the major concern of this study, the replaned operation of the tide gate at the Haimen Bay (HMB). The tide gate was firstly built in 1970 at the Lianjiang River in the southern part of China. It blocked the salty water inflow from the sea and overflow is only allowed when the water head in the upper stream is too high for the stock. Even though the local development has benefited from the tide gate's operation for the past several decades, the operation of the tide gate has resulted in a longer residual time of pollutants discharged by industries in the river upstream basin, which has caused critical levels of contamination [15]. To mitigate the severely polluted condition and help the ecological recovery, the reconstructed two-way tide gate at HMB to recover the tide receiving is expected to speed up the dispersion and advection of pollutants in the upstream. Besides the expected positive consequence, some negative impacts are also foreseeable, for example, increasing salty water intrusion and intensifying flushing of sediments [16]. Identification of the best practice to improve water quality as well as avoid saltwater intrusion and mitigate possible undesirable influence in the downstream part, however, is still a matter of further investigation and quantitative analysis.

To what extent the recovered tide-receiving could influence the pollutants' concentration and export flux merits quantization and concrete plan for environmental initiatives. Particularly for tide-influenced rivers and estuaries, tidal characteristics play an important role in the transport and concentration of sediments and waterborne materials [17,18]. As a typical tide-influenced river, the hydrodynamic transport in the Lianjiang River consists of processes characterized by different time scales. On the time scale of hours, floods and ebbs of semi-diurnal tidal cycles induce back–forth flow which results in periodical intrusion of seawater and flushing out of river water mass. The daily tidal cycle is further superposed by a fortnightly change of tide with a period of around half a month. Furthermore, other physical factors, such as precipitation rate, wind, river discharge varying in seasonal range in the Lianjiang Basin, will further complicate the situation. The tide flow, with interacting river runoff, determines the intrusion of salinity, resuspension of particulate matter, and dilution of pollutants.

In this study, based on the validated EFDC setup designed for the Lianjiang River, a scenario with tide-receiving (tide gate is kept open for the whole simulation period) was compared to a scenario without tide-receiving (i.e., the tide gate is open during the ebb tide, as historically planned). Simulation time was arranged in the dry season when the river is normally at a greater risk of pollution. Ammonia nitrogen was selected to represent the transport and distribution of pollutants. Besides the quantitative assessment of the improvements in the water quality, mechanisms accounting for the spatial and temporal variability of nitrogen's distribution is further investigated. The specific questions of

this study plan to resolve are: (1) quantifying the improvements in water quality, spatially and temporally, resulted from the tide receiving. (2) Pointing out the difference in time series of currents and nitrogen, modulated by the operation of the tide gate. (3) Quantifying the difference driven by the spring–neap cycle.

In the following sections, we first introduce the local natural and social characteristics of the Lianjiang River. Second, we provided the setting of the numerical model, the design of scenarios, and analyzing the routine of results. Third, in the result part, after the validation of the model setup, the variations of water quality in different scenarios were laid out and discussed. In the discussion part, we further explored the mechanisms responsible for the dispersion and dilution processes in the downstream part, in the context of hydrodynamic characteristics. The planning of tide receiving and its consequences were additionally discussed based on varied practical conditions and environmental criteria.

## 2. Materials and Methods

### 2.1. Study Site

Lianjiang River is one of the major rivers which flow into the South China Sea on the coast of eastern Guangdong province. The catchment area of the Lianjiang River is 1353 km$^2$, with the center of the basin located at 116.33° E, 23.26° N. The total population within this area is 4.67 million. The mainstream, with a length of 71 km, connects to the sea at its southern end the breakwater (Figure 1) and extends inland at the Puning city at 166.15° E, 23.36° N. The annual average runoff of the basin is 1.353 billion m$^3$ and the average annual rainfall is 1618 mm. The runoff and rainfall are unevenly distributed throughout the year, with less amount in winter and spring and more in summer and autumn. The seasonal variation in rainfall is related to the typical Southeast Asian monsoon climate. The southeast wind prevails in summer and northwest wind dominates in winter. The average wind speed for multiple years is 2.1 m s$^{-1}$, with a maximum wind speed of about 26 m s$^{-1}$. For the HMB, in addition to the fluvial influence, the impact of the ocean could further complicate the physical or even biological conditions. Currents flow along the coast of the Guangdong province, in the south-western to north-western direction, may add the influence of river runoff of the Pearl River (Figure 1) and nutrients input along the coast [19–21]. Driven by the south-western monsoon in summer, the Ekman transport tends to bring the surface water away from the coast. Therefore, the upwelling system adds a water body with high nutrients, high salinity, and low dissolved oxygen and temperature to the coastal water in the concerned area [22–24].

The tide gate at the HMB is a large-scale water conservancy project that was initially designed to prevent saltwater intrusion and improve the storage of freshwater. The tide gate is located near Haimen Town at the estuary of Lianjiang River, which is 2 km away from the river mouth (Figure 1). Since the 1970s, the operation of this tide gate has basically reduced the salty water intrusion caused by typhoons on more than 300,000 acres of farmland. It has effectively improved the irrigation of 187,000 acres of farmland on both sides of the Lianjiang River and guaranteed the water supply of living and industry use for more than 1,000,000 people.

Even though the local development has benefited from the tide gate's operation, the wetland ecosystems of the Lianjiang Estuary have been significantly changed. Since the 1990s, due to the population growth, the development of the textile printing and dyeing industry along the river, the pollutant loading has exceeded the ecological capacity [25]. The nutrients' concentration has long been below the inferior class V according to the Chinese Environmental Quality Standard for Surface Water (GB3838-2002; chemical oxygen demand (COD) > 40 mg L$^{-1}$, NH$_4$ > 2 mg L$^{-1}$, total phosphate (TP) > 0.4 mg L$^{-1}$, dissolved oxygen (DO) < 2 mg L$^{-1}$). In the lower reaches of the Lianjiang River, the water quality has not met the demand for production and living water in fisheries for a long time. The Lianjiang Estuary wetland ecosystem has almost lost its ecological service functions.

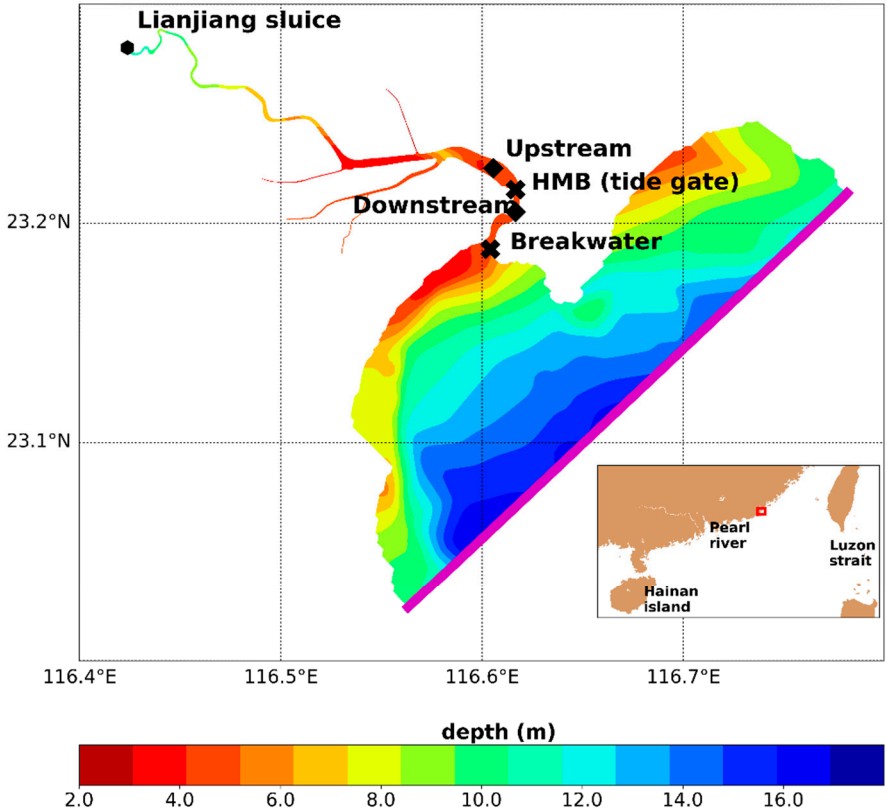

**Figure 1.** Bathymetry of the simulation domain. Cross-sections used for time series analysis are marked out by black diamonds and for flux calculations by black crosses. Open boundaries of upper and lower limits are depicted by a black dot and a magenta line, respectively. The inset shows the northern part of the South China Sea and the Lianjiang Basin which is enclosed by the red rectangle.

The reconstruction of the tide gate began in October of 2018 and planned to be complete in May of 2021. Before the restoration, the tide gate was planned to open during the ebb–tidal phase for draining. After the ebb tide, the aquifer area in the upper stream is refilled with runoff. The draining is more frequent in flood season than that in drought season. The specific operation of the tide gate has been determined by many anthropogenic factors, such as reservoir operation in the upstream, water surface level in the upstream, and demands for sewage draining. After the restoration, the tide gate is proposed to be kept open for the whole dry season when the pollution risk is severe and more dilution and dispersion are demanded.

*2.2. Numerical Model*

The EFDC model, originally developed at the Virginia Institute of Marine Science and later sponsored by the US Environmental Protection Agency (USEPA), is a modeling package for simulating 2D or 3D hydrodynamic fields coupled with several biogeochemical processes. Multiple modules are responsible to resolve varied components in the water environment, such as sediments and benthic substances, toxic contaminants, microalgae growth, and nutrient cycling. It has been applied worldwide in various surface water systems, such as rivers [26], lakes [27], estuaries [28], wetlands, and coastal regions [29]. The model solves the vertically hydrostatic, free-surface, turbulent averaged equations of motion for a variable-density fluid, using second-order accurate spatial finite-difference on a staggered C grid. Vertically, the sigma vertical coordinates are applied. The Mellor–Yamada level 2.5 turbulence closure scheme [30] is used to calculate turbulence parameters. Details of governing equations and numerical schemes for the EFDC hydrodynamic model are given by [31]. The code version 7.1 of EFDC was applied in this study. To simulate structural obstacles which blocks flow

across specified cell faces, in the EFDC model, the "mask.inp" function is designed to insert thin barriers which have widths much less than cell size or one grid spacing. Particularly, in this study, to implement the tide gate operation which is varying with time, the "mask.inp" was modified to play the "blocking" role in appointed time steps.

*2.3. Model Setting*

The simulation domain covers nearly a 30 km segment of the downstream part of the Lianjiang River (Figure 1). To the north, the open boundary of the upstream reaches a Lianjiang sluice in the mainstream, which is marked out in black dots in Figure 1. To the south, it includes the adjacent coastal area where the bay connects to the open sea. The magenta line represents the open boundary of the seaside (Figure 1). Along the river, several major tributaries have been resolved (Figure 1), with additional runoff and ammonia loading. Locations of the tide gate at the HMB and 2 other representative cross-sections, namely, the upstream cross-section relative to the HMB (UH) and the downstream cross-section relative to the HMB (DH), are marked out in black diamonds in Figure 1. Depicted by black crosses, the cross-sections of HMB and breakwater are used for flux calculation for the downstream portion (Figure 1). The model was formulated on orthogonal curve coordinate grids, with a spatial resolution ranging from 100 to 300 m. In total, it has 6264 simulation cells in the horizontal direction, with 5 layers in the vertical direction in the sigma coordinates. The model time step was 2.5 s and the output time step is 20 min, which allows for a robust representation of processes of physics and biogeochemistry. In the water quality module, dissolved oxygen (DO), total phosphate (TP), ammonia nitrogen ($NH_4$), and chemical oxygen demand (COD) were simulated. The benthic flux of ammonia is 14.4 mg m$^{-2}$ day$^{-1}$ [32,33] Nitrification and denitrification rates are set to be 0.1 day$^{-1}$, 0.09 day$^{-1}$ [34]. The manning coefficient is set as 0.02.

In the setup, necessary boundary and initial conditions were prescribed, including bathymetric information, meteorological conditions, loading of ammonia, and operation of the tide gate. The bathymetry in the Lianjiang River was measured in 2014 by the Shantou Hydrological Branch of Guangdong Hydrographic Bureau. The topographic of the study area is interpolated results of the nautical charts. The meteorological condition was provided by the Chinese Meteorological Science Data Sharing Service Network (http://data.cma.cn/). The data were originally measured at the meteorological station (116.58° E, 23.27° N) near the Lianjiang River. Among the measured meteorological data, wind, temperature, relative humidity, and water vapor pressure were implemented in the setup. The boundary of the study area was driven by the water elevation obtained from the Global Tide Assimilation Data (OTIS) [35], which was developed by Oregon State University in the United States. Sea surface elevation was prescribed at the open boundary of the with a time step of 1 h, which was sufficient to represent daily and half-month cycles. The eight most important tidal constituents ($M_2$, $S_2$, $N_2$, $K_2$, $K_1$, $O_1$, $P_1$, $Q_1$) are considered in the boundary forcing. Principal lunar ($M_2$) and principal solar ($S_2$) tides are responsible for most of the sea level variations in the semi-diurnal period, superposed by the fortnightly modulation due to the beating of the $M_2$, $S_2$. Relieving of diurnal inequality of tides during the neap phase has also been resolved properly. The water quality module and the field of salinity were initiated by the on-site water quality monitoring data of the area. Pollutants from tributaries were implemented as point sources entering the mainstream. By taking account of the distribution of sewage treatment plants and administrative divisions of sewage outlets, corresponding fluxes of pollutants were generalized into constant values for each tributary to represent spatial patterns of pollutants loading, particularly for the dry season. The full simulation period encompasses 35 days, which was initialized on 1 January, 2020 and ended on 5 February, 2020. The selected period is representative of the dry season of the Lianjiang River, when the polluted condition is generally more serious and water quality risk is higher than that in the wet season. For the validation, the simulation is forced with observed river runoff and ammonia concentration in January of 2020. The river runoff during that period fluctuated between 5–13 m$^3$ s$^{-1}$. Ammonia concentration was in the range of 4–9 mg L$^{-1}$. For the scenario analysis, in order to focus on the difference in ammonia

concentration and hydrodynamic processes that are caused by the tide gate's operation, the river runoff, and ammonia loading were assigned with a constant value. The runoff is 8 m$^3$ s$^{-1}$ and the ammonia concentration of the mainstream is 2 mg L$^{-1}$. Together with inputs from tributaries, loading of ammonia is 137.73 g s$^{-1}$ in scenario runs, which represents the expected condition after environmental measures are fully implemented. In reality, the operation of the tide gate is decided considering many factors besides the tidal phase. However, in the scenario runs, the opening of the tide gate in the non-tide-receiving scenario is arranged during the ebb phase in the dominating high tide, suggested by the Shantou Water Affairs Bureau.

Two scenarios were designed to evaluate the impacts of tide receiving. One was run with the tide gate opening for the whole period, which is called the tide-receiving scenario. In the other scenario, the tide gate was operated during the ebb–tidal phases, which was called the non-tide-receiving scenario.

*2.4. Analysis Methods*

Following the method proposed by Lerczak et al. and Zhou et al. [36,37], the transport of ammonia is decomposed into a seaward advection due to net outflow, estuarine transport with seaward flow at surface and landward flow near the bottom, and tidal oscillation.

$$N_s \approx \langle \int (u_0 N_0 + u_e N_e + u_t N_t) dA \rangle = Q_f N_0 + F_e + F_t \tag{1}$$

The component with subscript t varies predominantly on tidal scales. While components with subscript 0 and ε vary on subtidal time scales. Angled brackets indicate a low-pass, subtidal temporal filter, of which the details are described in [38].

To quantify the extent to which the water column is stratified, the stratification index (*N*) is calculated based on the difference in salinity between bottom and surface:

$$N = \frac{\delta S}{S_0} \tag{2}$$

where the $S_0$ is the vertically averaged salinity and the $\delta S$ is the difference in salinity between surface and bottom. The water column is mixed when $N < 0.1$. When *N* is in the range of 0.1–1, the water column is determined as partially mixed. When the *N* is higher than 1, there is evident stratification resulted from salty water intrusion.

Model results were evaluated quantitatively using 2 skill metrics, namely correlation coefficient (r) and simulation skills (SS) defined by Warner et al. [39]. In Equations (3) and (4), X is the variable and $\overline{X}$ is the corresponding mean value. Better agreement between model results and observations will yield a skill approaching one, and vice versa.

$$r = \frac{\sum_{i=1}^{N} (X_{mod} - \overline{X_{mod}})(X_{obs} - \overline{X_{obs}})}{\left[\sum_{i=1}^{N} (X_{mod} - \overline{X_{mod}})^2 \sum_{i=1}^{N} (X_{obs} - \overline{X_{obs}})^2\right]^{1/2}} \tag{3}$$

$$SS = 1 - \frac{\sum_{i=1}^{N} |X_{mod} - X_{obs}|^2}{\sum_{i=1}^{N} (|X_{mod} - \overline{X_{mod}}| + |X_{obs} - \overline{X_{obs}}|)} \tag{4}$$

## 3. Results

*3.1. Validation*

Surface elevation has been sampled since November 2019, in two monitoring stations located 200 m upstream and downstream to the HMB tide gate, respectively, with a temporal interval of 4 h. In the upstream station, ammonia has also been monitored. After data quality control and checking for data continuity of input variables, such as the record of tide gate operation, ammonia,

and runoff in upstream, observed data from 4–7th and 24–28th in January were selected to represent hydrodynamic and biochemical conditions in neap/spring tidal phase. The result of the comparison is shown in Figure 2 and quantified model skill results are listed in Table 1.

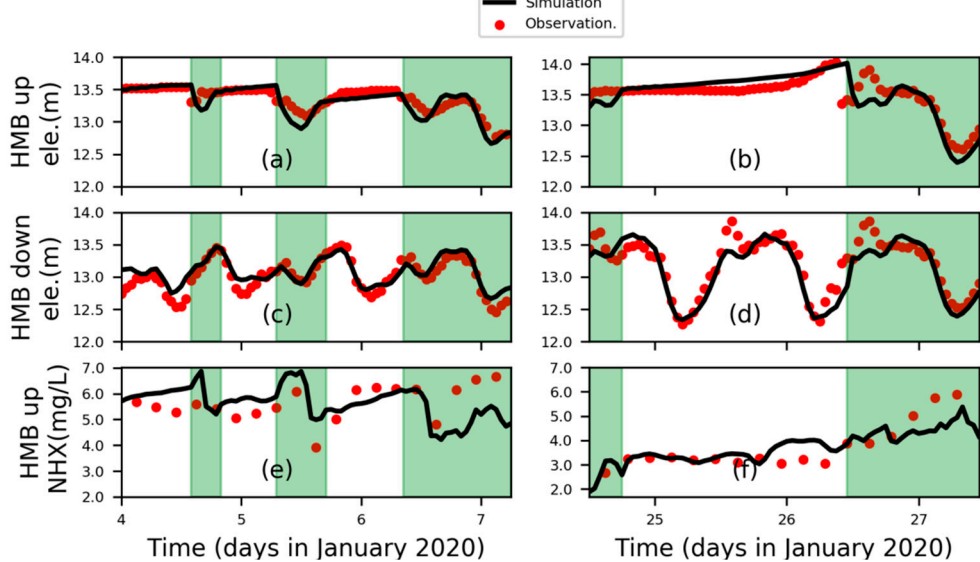

**Figure 2.** Comparison between simulation results and observed data. Water surface elevation in the upstream (**a**,**b**) and downstream (**c**,**d**) part of the HMB tide gate were plotted in the first and second row. Fluctuations of ammonia were shown in the last row (**e**,**f**). Time series of neap and spring phases were arranged in the left and right panel, respectively.

**Table 1.** Statistical summary of the difference between observation and results of the simulation, using correlation coefficient (r) and model skills (SS).

| Variable | Station | Time | | | |
|----------|---------|:----:|:--:|:----:|:--:|
| | | 2020.1.4–1.7 | | 2020.1.24–1.27 | |
| | | r | SS | r | SS |
| Elevation | HMB up | 0.87 | 0.96 | 0.89 | 0.95 |
| Elevation | HMB down | 0.85 | 0.93 | 0.90 | 0.95 |
| Ammonia | HMB up | 0.20 | 0.38 | 0.41 | 0.15 |

The simulation is able to capture the basic fluctuation of ammonia and surface elevation, which are influenced by the amount of input in the upstream and modulated by tide gate operation and propagation of tidal waves. When the tide gate was closed, the river runoff accumulated in the upstream of the HMB and water surface was gradually elevated (Figure 2a,b). When the tide gate was open, the tidal oscillation was detectable also in the upstream part. The ammonia in the upstream part also experienced an increase after tide receiving, mainly due to the transport of ammonia from the further upstream part where industrial and domestic sewage was discharged into the river. However, there are discrepancies in the ammonia concentration which mainly originated from boundary conditions. First, there were omissions in the records of operation of the tide gate. Even though these omissions did not happen in the time series which we used here, the influence on the transport and transformation of nitrogen in the whole simulation could not be excluded. Second, the loading of river runoff, nitrogen, and other contaminants from other tributaries was prescribed by constant values due to lacking continuous observations. This could contribute to the discrepancy of the definite amplitude of ammonia. However, the simulation is able to reflect the major variation in the hydrodynamic and biochemical processes, which depends on reasonable parameters' setting and reliable boundary forcing.

### 3.2. Variation of Ammonia Concentration

Given the improved sea–river connection under the identical inputs of ammonia load and river runoff, the concentration in the upstream part of the river has been reduced (Figure 3a–c). The descend range decreased gradually from upstream to downstream, from 0.64 mg L$^{-1}$ to 0.15 mg L$^{-1}$ (Figure 3c). The proportion of reduction shows the highest value in the downstream part near the breakwater (40%) (Figure 3d). The downstream portion relative to HMB experienced a relatively marginal reduction of 22% (Figure 3d). The most contaminated area is located in the north-eastern direction of the UD cross-section, with an ammonia concentration of 1.4 mg L$^{-1}$ in the tide-receiving scenario (Figure 3a) and 2.0 mg L$^{-1}$ in the non-tide-receiving scenario (Figure 3b). The portion within around 1 km north to HMB, enveloped by the contour line of 1.0 mg L$^{-1}$ and 1.4 mg L$^{-1}$ in the tide-receiving (Figure 3a) and non-tide receiving (Figure 3b), indicates an inburst of water intrusion from HMB, which results in a dilution of ammonia concentration. However, in the downstream portion relative to the HMB, the reduction of ammonia is limited to less than 0.3 mg L$^{-1}$ (Figure 3c). The strong tidal mixing homogenizes the spatial variation of reduction in the downstream. Nevertheless, the closer to the river mouth, the less amplitude of reduction is displayed, decreasing from 0.3 mg L$^{-1}$ close to the HMB to 0.15 mg L$^{-1}$ near the breakwater (Figure 3c).

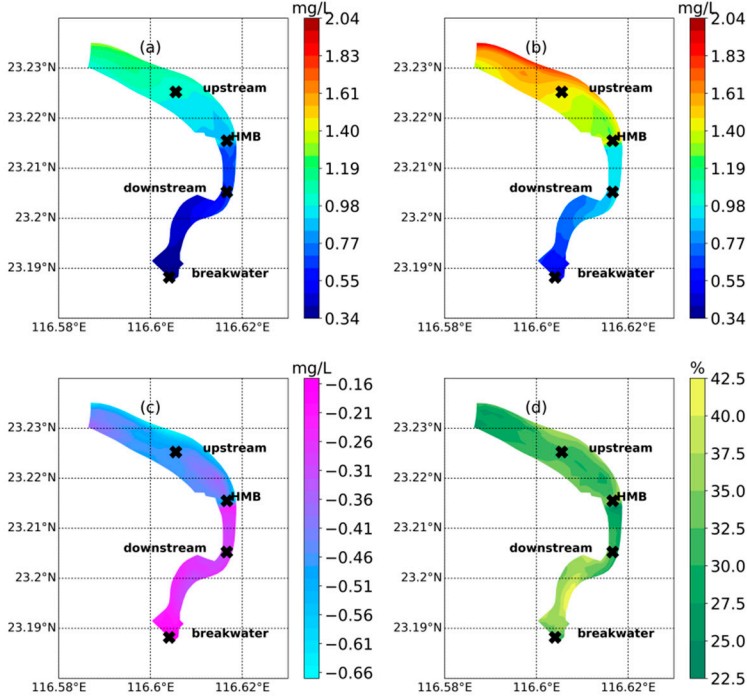

**Figure 3.** The averaged ammonia concentration for a spring–neap tidal cycle in the tide-receiving scenario (**a**), non-tide-receiving scenario (**b**), and the comparison between the two scenarios (tide-receiving scenario- non-tide-receiving scenario) (**c**). The relative change proportion is plotted in (**d**) (tide-receiving scenario minus non-tide-receiving scenario)/non-tide-receiving scenario).

### 3.3. Variations in Time Series

As shown in Figure 3, the reduction in ammonia shows different amplitudes and spatial variabilities. To further resolve the key processes responsible for spatial and even temporal variability, time series of three typical cross-sections, which represent the upstream portion (UH), the downstream portion (DH) relative to HMB, and the HMB tide gate were selected for further investigations of the time series related to tidal cycle and the operation of the tide gate (Figure 4). The time series of averaged concentration of ammonia and flux balance in the downstream part (from HMB to breakwater) are resolved in Figure 5. The spatial extent and duration of variations in ammonia concentration caused by tide-receiving are

revealed in a Hovmöller diagram built along the downstream part of the river, covering spring–neap cycles (Figure 6). In Figure 6, data have been weighted averaged in each cross-section along the river. The spatial upper limit of this diagram is the last tributary close to the river mouth.

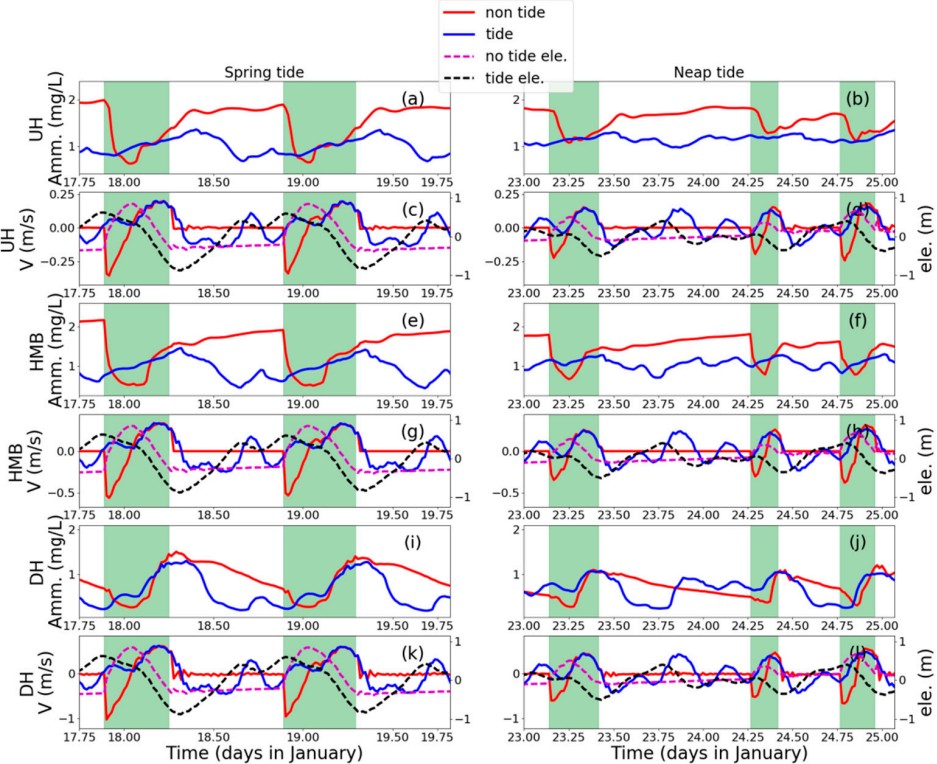

**Figure 4.** Time series of velocity and ammonia concentration in 3 cross-sections: downstream part relative to HMB (UH) (**a**–**d**), HMB tide gate) (**e**–**h**), and downstream part relative to HMB (DH) (**i**–**l**) (Figure 1). Two semi-diurnal cycles, in the spring phase (17.75th–19.75th of January) and neap phases (23rd–26nd of January), were plotted on the left panels and right panels. Values in the tide-receiving scenario were plotted in blue curves and the non-tide-receiving scenario in red curves, with tide gate opening periods marked out by green shadows. Tidal elevations of HMB were plotted together with velocity to show the tidal phases (c, d, g, h, k, l). For velocity, seaward is a positive value and vice versa.

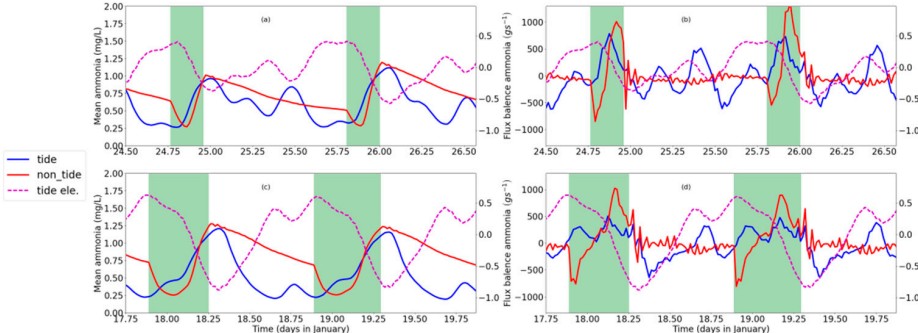

**Figure 5.** The averaged concentration of ammonia in the cross-section between the HMB (tide gate) and breakwater in the neap (**a**) and spring phase (**c**). The balance of ammonia flux (seaward flux from HMB minus seaward flux from breakwater) during the neap (**b**) and spring (**d**) phase. Seaward transportation is a positive value. Concentrations and flux of ammonia in the tide-receiving scenario are in blue and non-tidal scenario in red. Water surface elevation is also plotted in dash lines in magenta to indicate the tidal phase. The tide gate opening period is marked out by the green shadow.

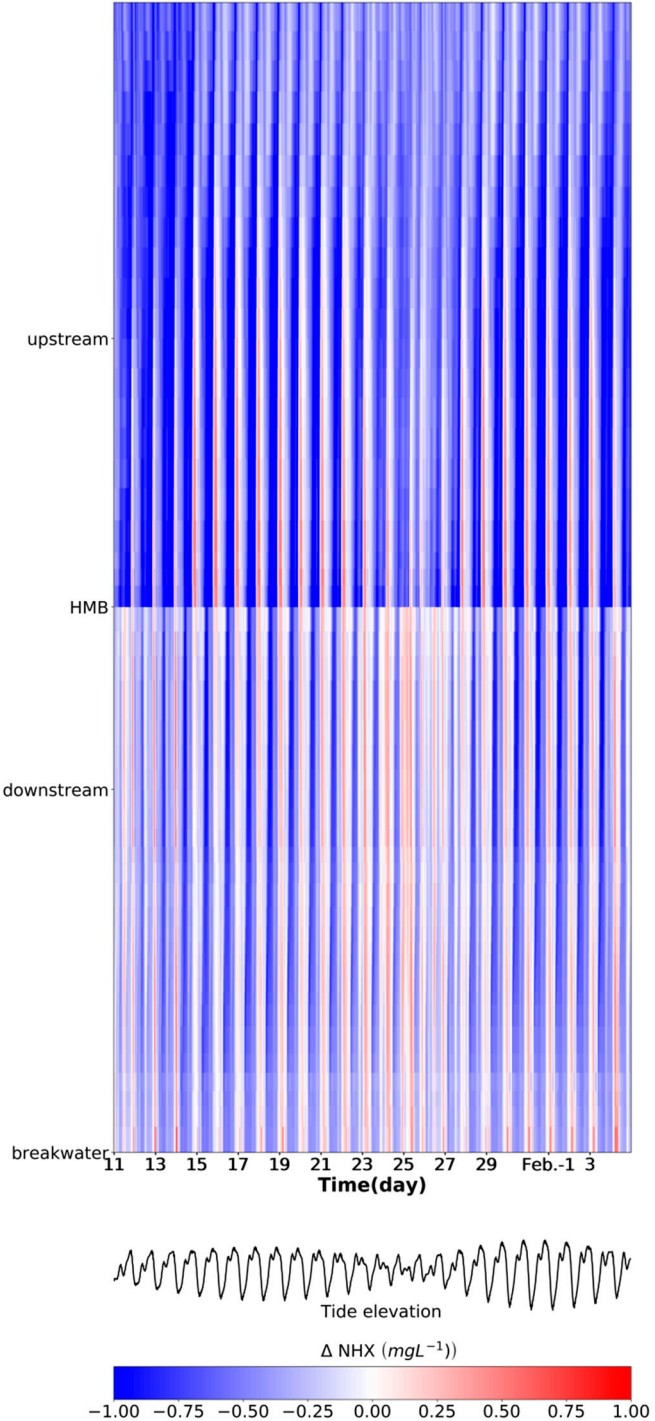

**Figure 6.** Hovmöller diagram of the difference (tide-receiving scenario minus non-tide-receiving scenario) in the concentration of ammonia along the river. The *x*-axis represents the time evolution and the *y*-axis depicts the location along the river. The tide elevation is plotted in the lower panel to identify tidal phases.

The opening of the tide gate in the non-tide-receiving scenario is arranged in the falling of high-high water in each daily cycle. In the non-tide-receiving scenario, when the tide gate is closed, water bodies in the upstream part is blocked from the downstream and experiences gradually increasing of the ammonia value (Figure 4a,b,e,f). After the tide gate is opened, the concentration in the upstream increases and drops down in the first half of the opening period and then increases

again. The concentration returns to the prior level after the closing of the tide gate in 2–3 h in the neap phase (Figure 4b,f) and in around 6 h in the spring phase (Figure 4a,e). Without the modulation of the tide gate, in the tide-receiving scenario, the water body mainly fluctuates with tidal velocities, which brings the water body with high ammonia value seaward in the ebb phases and transport water body from downstream with lower concentration landward during flood phases. The tide- or tide-gate-regulated oscillation in ammonia concentration is more evident in the spring phase (Figure 4a,e,i and Figure 5c) than in the neap tide (Figure 4b,f,j and Figure 5a).

When the tide gate is opened in the non-tide-receiving scenario during the spring phase, the water body floods landward from the tide gate in the HMB (Figure 4g) and the ammonia concentration decreases from 2.2 mg L$^{-1}$ to around 0.5 mg L$^{-1}$ in 2 h (Figure 4e). In the later period of the opening-gate operation, the HMB experiences an increase in the ammonia concentration, which is resulted from seaward transport of the polluted water body from the further upstream part. The landward intrusion in the first half of the tide opening period is not only predominant in the upstream part relative to the HMB, but also in the downstream part (Figure 4k). The negative flux of ammonia (landward) in the first half of the tide gate opening period and positive flux (seaward) in the second half of the tide gate opening period are also visible in the flux balance in the downstream portion (Figure 5b,d).

Compared to the regular and obvious change along tidal cycle in the spring phase (Figure 4e,g), during the neap phase, the asymmetry of the semi-diurnal tide in a lunar day decreases, which results in less distinction between higher-high tide, lower-high tide and associated tidal currents (Figure 4d,h,l). The basic variation pattern, i.e., a decrease in ammonia concentration due to landward intrusion of seawater and an increase in ammonia after the currents shifts seawater still holds for the neap phase. However, because of the comparably weaker seawater intrusion in the neap tide, the amplitude of response in ammonia concentration is smaller in neap tide (no more than 0.5 mg L$^{-1}$ decreased by tidal cycle and around 0.75 mg L$^{-1}$ by the operation of tide gate; Figure 4b,f,j and Figure 5a) than spring tide (around 1mg L$^{-1}$ driven by tidal cycle or the operation of tide gate; Figure 4a,e,i and Figure 5c).

Besides the similar pattern of the responses of velocity and ammonia concentration in the tide gate opening period, the most distinct pattern between the upstream and downstream parts in the non-tide scenario occurs when the tide gate is closed. At the end of the tide gate opening period, the ammonia concentration at the UB reached 1.75 mg L$^{-1}$, which continues its increase for 3 h after the close of the tide gate (Figure 4a,b,e,f). It maintains a high level of around 2.5 mg L$^{-1}$ until the next re-opening of the tide gate. In the downstream part, on the contrary to the situation of upstream which experienced an accumulation of ammonia in the tide gate closing period, the DH receives dense ammonia flux in the second half of the tide gate opening period and recover better water quality in the rest part of a tidal cycle (Figure 4i,j).

Comparing the ammonia concentration between two scenarios, we found that in most part of the time period, the water quality in the tide-receiving scenario is better than the non-tide-receiving scenario, which is in agreement with the averaged pattern for a spring–neap cycle (Figure 3c). However, there are some exceptional conditions when the water quality deteriorates after the tide receiving. In the UH site, the ammonia concentration is lower in the tide-receiving scenario except for the period when the reduction of ammonia concentration after the tide gate opening reaches the maximum (around 18 and 19 January) in the spring phase (Figure 4a,e). This phenomenon lasts for 2–3 h during the spring phase and could cover the spatial range from the breakwater to the UH section (Figure 6). This may be also associated with the lower averaged concentration spatially contoured by 1.13 mg L$^{-1}$ (Figure 3a) and 1.4 mg L$^{-1}$ (Figure 3b), which depicts the spatial coverage of the obvious landward intrusion of seawater. Compared to the spring phase, a weaker landward intrusion and consequential limited reduction of ammonia concentration in the neap phase around the UH cross-section (Figure 6). The reduction is limited and the ammonia concentration in the non-tide scenario stays higher in the further UH (Figure 4b). On the contrary, in the downstream part, during the neap tide, the duration time when the ammonia concentration in the tide-receiving scenario is higher could last for around half a day at the DH site (Figure 4j, 23.8–24.4 d, January) and other

downstream portions (Figure 6). The several peaks in the ammonia concentration time series coincide with the seaward flow (Figure 4l) and seaward flux of ammonia (Figure 5b) input from HMB, which indicates back–forth oscillation of water body with higher ammonia concentration in the neap tide.

## 4. Discussion

Concerning the ammonia flux compromised by different transport processes that are quantified by Equation (1), the tidal oscillation plays the dominant role and shows the spring–neap fluctuation. The instantaneous ammonia flux driven by the tide (140 Ton/spring–neap cycle) is around one order larger than that transported by river net-flow and estuarine circulation (15 Ton/spring–neap cycle, Figure 7c,d). Even though the tidal oscillation flux has been modulated by the operation of the tide gate, i.e., little landward flux when the tide gate is closed in the non-tide-receiving scenario (Figure 7b), for a spring–neap cycle, the accumulative fluxes are comparable between two scenarios with a decrease of around 8 Ton in the tide-receiving scenario. The tide oscillation transport is undermined during the neap phase, which can be verified by the moderate sloping of accumulative flux driven by tide during the neap phase (24–25th of January, Figure 7a,b). During the neap phase, in the non-tide-receiving scenario, the seaward transport of ammonia is intensified during the neap phase (Figure 7d), which is induced by the enhanced stratification (Figure 8). However, this is not detectable in the tide-receiving scenario.

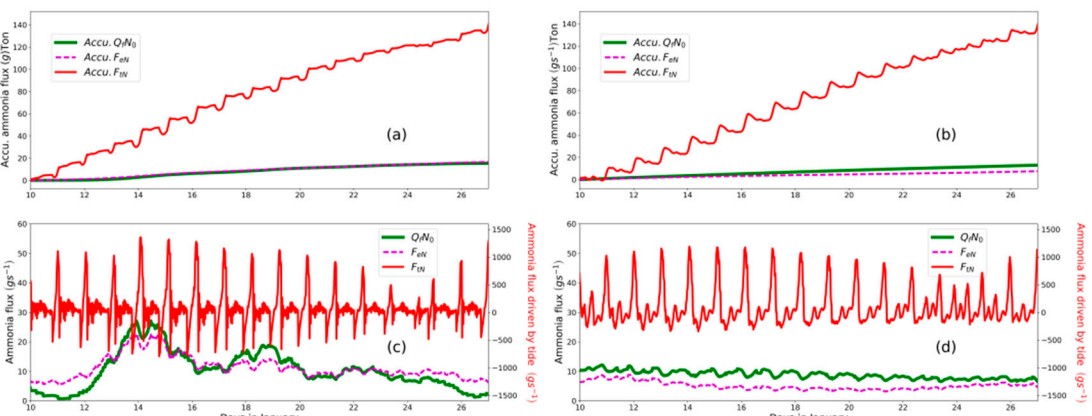

**Figure 7.** Ammonia flux resulting from tidal oscillation (red line), subtidal estuarine circulation (magenta dash line), net outflow (green line) in the tide-receiving scenario (**d**) and the non-tide-receiving scenario(**c**). Accumulative flux since the 10 January for the non-tide-receiving scenario is plotted in (**a**) and tide-receiving scenario in (**b**). A positive value indicates an oceanward flux.

The predominant contribution from tidal oscillation transport and elevated flux during spring phase are not well in line with the phenomenon observed in other estuary systems, such as Modaomen Estuary [37] and Hudson River Estuary [40], where salty water intrusion and material transport are more prevalent in the neap phase. In those cases, the stratification is further enhanced in the neap tide and hence aggravating salty water intrusion from the bottom landward and fluvial materials on the surface layer transport seaward [41]. In the Lianjiang River, the major part of the downstream portion is partially mixed, except for a short period during the ebb phase (Figure 8e–h). Given the limited river runoff, shallow and flat estuary shape and weak stratification, the transport of material is mainly driven by tide oscillation instead of net outflow and estuary circulation.That is also why the tidal frequency is predominant in the variation of ammonia concentration (Figure 4) and flux (Figure 5) in the estuary part of the river.

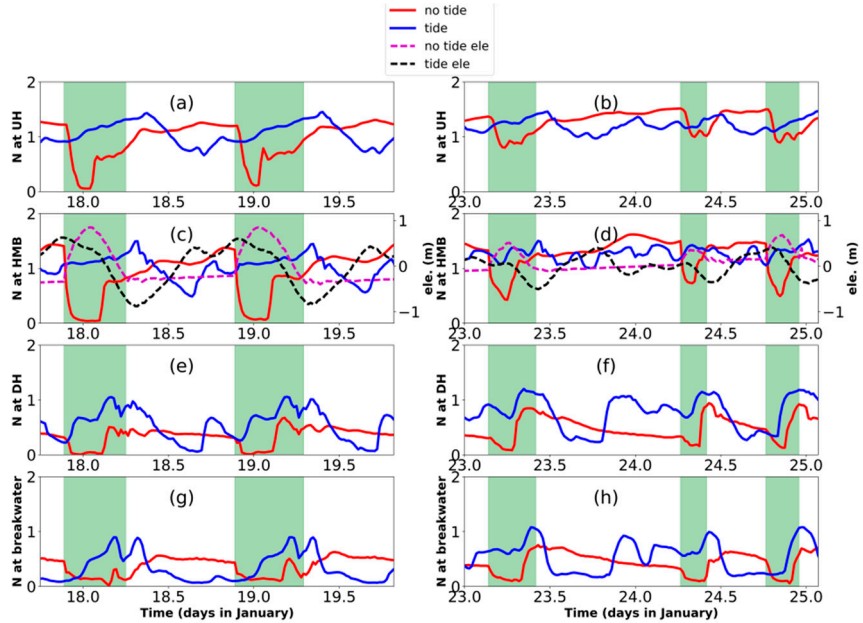

**Figure 8.** Time series of stratification index $N$ (calculated by Equation (2)) in 4 cross-sections: UH (**a**,**b**); HMB (**c**,**d**); DH (**e**,**f**) and breakwater (**g**,**h**). Spring (17.75th–19.92th of January) and neap phases (23rd–25.07th of January), which contain 2 semi-diurnal cycles, were plotted on the left panels and right panels. Values in the tide-receiving scenario were plotted in blue curves and in the non-tide-receiving scenario in red curves, with the tide gate opening period marked out by green shadows. Tidal elevations of HMB were plotted to show the tidal phases (**c**,**d**).

In this study, the distribution of ammonia was mainly attributed to physical transport and mixing. Biogeochemical processes, such as nitrification, denitrification [42], and consumption by algae [43], may also be responsible for the transform of ammonia [44], but account for limited proportion in this case. The limited biogeochemical transformation of ammonia can be verified by the relationship between salinity and ammonia (Figure 9). If a linear relationship exists between a dissolved constituent and salinity which is a conservative indicator of mixing between seawater and freshwater, then the dissolved constituent can be described as conservative in this mixing process. This assumption holds approximately in the tide-receiving scenario in the downstream portion relative to UH. The correlation between salinity and ammonia fluctuates around −0.75, with $p$ value < 0.01, indicating that removing or addition of ammonia is often a relatively minor process affecting the distribution compared to physical dilution. In both scenarios, the R value increases in the upstream portion relative to HMB. The curve for the tidal scenario also increases near the river mouth (breakwater), which indicates that the back-force oscillation caused by tide may result in longer residual time for ammonia near the river mouth and allow other biogeochemical processes to take part in. This is also suggested by the budget calculation that the net exported ammonia flux in the tide-receiving scenario is less than that in the non-tide-receiving scenario (Table 2). Furthermore, the correlation is not robust in the further upstream portion relative to the UH (data not shown), which indicates that the behavior of ammonia was subject to in-situ uptake or removal. It also suggests that the biogeochemical processes have to be taken into account if we aim at giving a further quantitative explanation of ammonia or even total nitrogen's budget with more in situ measurements.

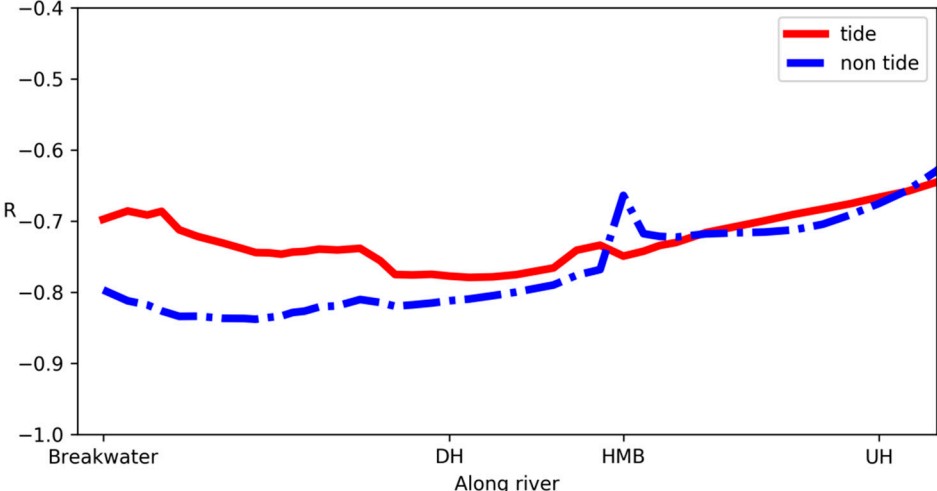

**Figure 9.** The correlation between salinity and ammonia nitrogen along the river from breakwater until the UH cross-section. The R values of the tide and non-tide-receiving scenarios are plotted in red and blue, respectively. *p* values are lower than 0.01.

**Table 2.** Seaward flux of ammonia and water mass at the cross-section of HMBand breakwater in both scenarios for a lunar cycle in the spring/neap phase and for a spring–neap cycle.

| Sections | Variable | Concerned Time Period | | Neap Tide | Non-Tide | Spring–Neap Cycle | |
| --- | --- | --- | --- | --- | --- | --- | --- |
| | | Spring Tide | Non-Tide | | | Tide | Non-Tide |
| HMB | Ammonia Flux Seaward | (Ton/Lunar Cycle) | | | | (Ton/Spring–Neap Cycle) | |
| | | 10.52 | 13.15 | 4.52 | 8.56 | 126.23 | 165.65 |
| | Breakwater | 10.85 | 10.45 | 6.74 | 7.93 | 135.52 | 144.46 |
| HMB | Flux Seaward | ($10^4$ m$^3$/Lunar Cycle) | | | | ($10^4$ m$^3$/spring–neap cycle) | |
| | | 196.77 | 379.86 | −40.08 | 361.46 | 12,623.16 | 16,565.13 |
| | Breakwater | 219.34 | 146.91 | −66.88 | 200.89 | 13,551.69 | 14,446.32 |

The consequences of different tide gate operations are mainly evaluated by the dilution of ammonia concentration. However, there are varieties of criteria for evaluations of the operation planning which would be considered in the future. With elevated water storage in the river in the tide-receiving scenario, the dense ammonia has been diluted. The decreased ammonia is expected to benefit the recovery of the ecological function of the Lianjiang River. Particularly in the dry season when the storage of water in the Lianjiang River is much lower than the necessary ecological water demand. The advantage with more available water storage is well beyond mere considering a decrease of pollutants. Even though the water storage has been elevated, due to the back–forth currents in the tide-receiving scenario, the total exported amount of ammonia decreased by 6% at the cross-section of the breakwater and 23% at the HMB (Table 2). The downsteam part relative to HMB experiences higher concentration during neap tides. In this regard, the advantage of the tide-receiving scenario, with no control over the natural flow, has to be reconsidered, in aspects of the flux of total nitrogen, residual time of pollutants and so on. It is worth noting that the increase of ammonia concentration, or longer duration of higher-level ammonia in the downstream relative to the HMB may influence aquatic agriculture. In the future study, according to varied standards or targets of environmental protection, a variety of evaluations should be further explored, such as residual time of pollutants, duration of high-level concentration of specific constituents, and export flux of contaminants.

As the first trial to evaluate the impact of modulation of the tide gate on ammonia transport, the operation plan of the tide gate has been simplified to design the non-tide-receiving scenario. For the environmental protection practice in Lianjiang, contingency plans or affirmative action would be taken depending on many factors. Even though the tide gate is mainly planned to open in the ebb phase, for exporting pollutants from the upstream, however, tide gate normally would not be switched on–off

every day affirmatively in reality during the dry season. In the dry season, due to the low runoff, the water level in the upstream portion relative to HMB is often lower than the water level downstream. That is why we see the prevalent intrusion of seawater after the tide gate opens. Conversely, abundant precipitation in the wet season results in more frequent openings of the tide gate, mainly for flood releases. Typhoon induced storm surge often happens in the adjacent coastal area of Lianjiang, for which the tide gate has to be closed to block flooding from the seaside. In addition, the hydraulic scheduling of reservoirs in the upstream will further complicate water storage and scheduling of the tide gate at the HMB. In case unexpected pollutants release happened, the tide gate would also be used for emergency response. Studies for practical cases are possible and will be more valuable after the completion of the reconstruction of the tide gate and more in-situ data are available.

## 5. Conclusions

To improve the environmental health in the Lianjiang River, recovery of tide receiving has been proposed to reinforce the connection between coastal area and river. This study is the initial attempt to evaluate the variation induced by tide-receiving, mainly in terms of the concentration and flux of ammonia, with a focus on the dry season when the environmental risk is higher. Compared to the non-tide-receiving scenario which blocks the currents in the flood phase, the tide-receiving scenario with no modulation on natural tidal currents allows more water mass from the seaside to flood in and reduce the ammonia concentration by 20–40%. With tide receiving, the concentration of ammonia fluctuates with the major tidal cycles, increasing in the ebb phase when the polluted water mass in the further upstream are transported seaward and decreasing in flood phase when seawater intrudes to upstream. The tidal oscillation has been modulated with the operation of the tide gate. The landward intrusion occurs right after the opening of the tide gate, followed by the seaward transport of upstream water and an increase of ammonia concentration. The extent of landward intrusion and decreasing of ammonia concentration in the upstream are intensified in the spring phase rather than the neap phase. However, the unmodulated back–forth tidal currents in the tide-receiving scenario end up with less export flux of ammonia.

Both natural characteristics and anthropogenic influence should be considered in the initiative of environmental protection. The results reveal the dominating role played by the tide in the transport of waterborne material in the Lianjiang River, due to the mixing environment in the estuary. Physical mixing and transport are mainly responsible for the distribution of ammonia. Even though the concentration has decreased in the tide-receiving scenario, other aspects of the impact, such as duration time of high concentration for specific segments in the river, merits further investigation in the future.

**Author Contributions:** Conceptualization, Z.F. and C.Z.; methodology, L.Z. and C.Z.; setting up of the simulation, H.Y. and W.W.; validation, C.Z. and H.Y.; analysis, C.Z., Z.F., W.W. and L.Z.; writing—original draft preparation, C.Z.; writing—review and editing, Z.F. and L.Z.; visualization, C.Z.; supervision, F.Z.; project administration, F.Z. and Z.F.; funding acquisition, F.Z. All authors have read and agreed to the published version of the manuscript.

**Funding:** This work is funded by the Key-Area Research and Development Program of Guangdong Province (No. 2019110205003) and the Basal Specific Research of the Central Public-Interest Scientific Institute (Grant No. PM-zx097-202002-069, Grant No.PM-zx703-202004-141).

**Acknowledgments:** All authors are indebted to the Environmental Protection Monitoring Station of Shantou City for providing the observed data. Authors are grateful to Gang Chen and Zhong Wang for managing datasets.

**Conflicts of Interest:** The authors declare that they have no known competing financial interests or personal relationships that could have appeared to influence the work reported in this paper.

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
