# Peer review of "Impacts of Tide Gate Modulation on Ammonia Transport in a Semi-closed Estuary during the Dry Season—A Case Study at the Lianjiang River in South China"

_water, doi:10.3390/w12071945_

Round 1
Reviewer 1 Report
Page 1 line 23; Page 4 line 129 (mm per m2), line 133 i 134 (m·s-1), line 154 i 155 (mg·L-1…); Page 6 line 224, 225, 228; Page 6 line 229 (Ton per day). General please standardize all units in article
Page 3 line 99; Page 4 line 126, 129, 144 space in place
Page 3 line 111 Please change “study site” into Study sites
The signature in Figure 1 is too long, shorten all signatures in article. Details should be in the materials and methods section.
Page 4 line 129 please change index m3
Page 4, line 154, 155 Please record should be amended
Page 5 line 194 Please correct the entry for TP (total phosphate), ammonia nitrogen (NH4)
The Authors first introduced the local natural and social characteristics of the Lianjiang River. In study results saw the setting of the numerical model, the design of scenarios and analyzing of results. In the result part, after the validation of the model setup, the variations of water quality in different scenarios were laid out and discussed. In the discussion part, they further explored the mechanisms responsible for the dispersion and dilution processes in the downstream part, in the context of hydrodynamic characteristics. The Authors present the planning of tide receiving and its consequences were additionally discussed based on varied practical conditions and environmental criteria.
The work is written concisely, it does presents a problem that has arisen in the world and is part of the subject matter of the journal.
Author Response
Dear reviewer:
We highly appreciate your constructive comments. These greatly helped to improve the manuscript. Point-to-point revision notes are provided in the following. For convenience, we also provide line numbers (corresponds to the revised version without tracking changes) where we have changed the contents.
- Page 1 line 23; Page 4 line 129 (mm per m2), line 133 i 134 (m·s-1), line 154 i 155 (mg·L-1…); Page 6 line 224, 225, 228; Page 6 line 229 (Ton per day). General please standardize all units in article
Author response: Yes, according to your suggestions, we have standarded all units. We have changed mg/L to mg L-1(Page 1, line 23) 1618mm/m2 to 1.618m m-2(page 4, line 131), m/s to m s-1(page 4, line 135, 136), mg/L to mg L-1 (page 4, line 156,line 157), m3/s to m3s-1 (page 6, line 234,line 237), mg/L to mg L-1 (page 6, line 235,line 238),119Ton/day to 137.73gs-1(page 6, line 239).
- Page 3 line 99; Page 4 line 126, 129, 144 space in place
Author response: Yes, we have added spaces in page 3 line 100, 101,102. In page 4, we have standarded the units of locations, in line 127, to keep alignment with line 129. We have added spaces in line 129, 127.
- Page 3 line 111 Please change “study site” into Study sites
Author response: yes, we agree. We changed the study site into Study sites in line 122.
- The signature in Figure 1 is too long, shorten all signatures in article. Details should be in the materials and methods section.
Author response: Yes, we agree. We have shortend the caption of Figure.1 in line 117 to line 121. We put details in the model setting section, in line 243 to line 250. For figure.5, the caption has also been shortend and details were rewritten in line 391 to line 392.
- Page 4 line 129 please change index m3
Author response: Yes, we agree. We have made the ‘3’ as superscription and now it is in line 127
- Page 4, line 154, 155 Please record should be amended
Author response: Yes, we have checked the GB3838-2002 again and corrected our errors in line 152-153. (COD>40 mgL-1, NH4>2 mgL-1, TP >0.4 mgL-1, DO < 2 mgL-1)
- Page 5 line 194 Please correct the entry for TP (total phosphate), ammonia nitrogen (NH4)
Author response: Yes, we have corrected this error in line 251.

Reviewer 2 Report
The authors of this manuscript use models to evaluate the impacts of tide-gate modulation on the water quality in the Lianjiang River, connected to the South China Sea. The authors use ammonia transport as a marker for processes up- and downstream the tide gate, in the dry season, when particularly relevant for its ecological impact on the region.
The manuscript’s data are a good starting point and of preliminary value for a suite of more expansive modeling of other aquatic and biological processes in the area, relevant for the sustainability of tide regulating approaches and ecological protection of the area.
The manuscript is generally well written, and sound, but there are several typos in the text, missing spaces, capital letters not used where needed, and inconsistency in the use of SI units. The manuscript would benefit from a thorough read-through. Some examples: rows 99-101; row 111 – 2.1 Study site needs capitalized; row 129 – m3 use superscript; rows 155-156 – correct the sentence “ …the water quality has been not met the demand for production…”using “ the water quality did not meet the demand for production”; row 230 – sentence “in reality…” capitalize; row 238 – 2.4 “Analyzing Methods” replace with “Analysis Methods”; row 314 – 3.3 “variations in time series” capitalize; row 333 – “The Opening” replace with “The opening”; row 337 – reformulate sentence to “…in the first half of the tide-opening period”; row 399 – Fig.6 description, capitalize; row 446 – “…may also responsible for the transform of ammonia but account for limited..” correct to “…may also be responsible for the transformation of ammonia, but account for a limited…”. This statement also needs a reference.
Since the model only spans a short period of time, it would be beneficial for the authors to include an assessment on how longer-term variations may affect model output (possibly, a goal of a further study).
Author Response
Dear reviewer:
We highly appreciate your constructive comments. These greatly helped to improve the manuscript. Point-to-point revision notes are provided in the following. For convenience, we also provide line numbers (corresponds to the revised version without tracking changes) where we have changed the contents.
- The manuscript is generally well written, and sound, but there are several typos in the text, missing spaces, capital letters not used where needed, and inconsistency in the use of SI units. The manuscript would benefit from a thorough read-through.
Author respesponse: Yes. We have read through the whole manuscript again to eliminate typo errors.
- rows 99-101:
Author respesponse: We have added spaces after brackets in line 99-101.
- row 111 – 2.1 Study site needs capitalized;
Author response: the study site has been changed to Study sites in line 111
- row 129 – m3 use superscript;
Author response: we have superscripted the ‘3’ in line 131.
- rows 155-156 – correct the sentence “ …the water quality has been not met the demand for production…”using “ the water quality did not meet the demand for production”;
Author response: we found the sentence is confusing. There is no necessity to mention the demands of living water of people or use in fishery agriculture. It relates to classification of functional purposes of different areas, which is beyond the scope of this paper. To avoid further misunderstanding, we delete this sentence.
- row 230 – sentence “in reality…” capitalize;
Author response: we capitalized the “In” in line 240
- row 238 – 2.4 “Analyzing Methods” replace with “Analysis Methods”;
Author response: Yes, we agree. We have changed the Analyzing methods to Analysis Methods in line 248
- row 314 – 3.3 “variations in time series” capitalize;
Author response: Yes, we agree. It has been corrected as “Variations in time series” in line 325.
- row 333 – “The Opening” replace with “The opening”;
Author response: Yes, we agree.We have changed the ‘Opening’ to ‘opening’ in line 347.
- row 337 – reformulate sentence to “…in the first half of the tide-opening period”;
Author response: Yes, this sentence was confusing. We have rewritten it as ‘After the tide gate is open, the concentration in the upstream increases drops down in the first half of the opening period and then increase again.’
- row 399 – Fig.6 description, capitalize;
Author response: Yes, we agree. We have capitalize the first of letter of the first word of this caption in line 413
- row 446 – “…may also responsible for the transform of ammonia but account for limited..” correct to “…may also be responsible for the transformation of ammonia, but account for a limited…”. This statement also needs a reference.
Author response: Yes, we agree. We have added the verb ‘be’ in this sentence. The original sentence is misleading. We rewrite it as: ‘Biogeochemical processes , such as nitrification, denitrification (Romero et al., 2019), and consumption by algae (Peierls et al., 2012), may also be responsible for the transform of ammonia (Cloern et al., 2014), but account for limited proportion in this case.’ (line 459-461) We have added some references which studies the biogeochemical processes’ influence on nutrients’ fate and species. However, in this study, the contribution of biogeochemical processes is limited. We verified this phenomenon in this local case in following sentences, which are related to Fig.9.
- Since the model only spans a short period of time, it would be beneficial for the authors to include an assessment on how longer-term variations may affect model output (possibly, a goal of a further study).
Author response: Thank you for your nice suggestion. It is true that in this study, the validation is limited in a short period and the scenario analysis only represents a single condition. Unfortunately, as the first manuscript of the Lianjiang river and adjacent estuary, it is limited by many factors. First, the monitoring station at the Lianjiang Sluice at the upstream and the HMB tide gate was firstly established last November and has been run through testing. The stability of the equipment and data quality demands further checking before we further span our analyzing period and test our model for longer simulation. Second, as I have mentioned in section 3.1, omission in the records of operation of the tide gate is one of the major reason which hinders longer validation. The operation of the tide gate was recorded manually. Once the opening/closing is missing, we lose a reliable “boundary condition” for the simulation. I am also concerning the progress of the technical obstacles. Third, the major focus of this study is the dry season. In flood seasons, the purpose operation of the tide gate is different, which mainly aims at draining the river run-off after heavy rainfall. In that case, the pattern of the contaminants is different from that in dry seasons.
We also agree with you that water quality in a longer period and even extreme weather conditions merits further exploration and are planned in future studies. In the lower reaches of the Lianjiang River, the water quality has been not met the demand for production and living water in fisheries for a long time.
